# Fatty Acid Profiling as a Tool for Fostering the Traceability of the Halophyte Plant *Salicornia ramosissima* and Contributing to Its Nutritional Valorization

**DOI:** 10.3390/plants13040545

**Published:** 2024-02-16

**Authors:** Fernando Ricardo, Ana Carolina Veríssimo, Elisabete Maciel, Maria Rosário Domingues, Ricardo Calado

**Affiliations:** 1Laboratório para a Inovação e Sustentabilidade dos Recursos Biológicos Marinhos (ECOMARE), Centro de Estudos do Ambiente e do Mar (CESAM), Departamento de Biologia, Universidade de Aveiro, Campus Universitário de Santiago, 3810-193 Aveiro, Portugal; 2Centro de Estudos do Ambiente e do Mar (CESAM), Departamento de Química, Universidade de Aveiro, Campus Universitário de Santiago, 3810-193 Aveiro, Portugal; carolinaana@ua.pt (A.C.V.); elisabete.maciel@ua.pt (E.M.); 3Laboratório Associado para a Química Verde (LAQV-REQUIMTE), Departamento de Química, Universidade de Aveiro, Campus Universitário de Santiago, 3810-193 Aveiro, Portugal; 4Centro de Espetrometria de Massa, Laboratório Associado para a Química Verde (LAQV-REQUIMTE), Departamento de Química, Universidade de Aveiro, Campus Universitário de Santiago, 3810-193 Aveiro, Portugal

**Keywords:** GC-MS, geographic origin, healthy products, lipid markers, sea asparagus

## Abstract

*Salicornia ramosissima*, commonly known as glasswort or sea asparagus, is a halophyte plant cultivated for human consumption that is often referred to as a sea vegetable rich in health-promoting *n*-3 fatty acids (FAs). Yet, the effect of abiotic conditions, such as salinity and temperature, on the FA profile of *S. ramosissima* remains largely unknown. These factors can potentially shape its nutritional composition and yield unique fatty acid signatures that can reveal its geographical origin. In this context, samples of *S. ramosissima* were collected from four different locations along the coastline of mainland Portugal and their FAs were profiled through gas chromatography–mass spectrometry. The lipid extracts displayed a high content of essential FAs, such as 18:2*n*-6 and 18:3*n*-3. In addition to an epoxide fatty acid exclusively identified in samples from the Mondego estuary, the relative abundance of FAs varied between origin sites, revealing that FA profiles can be used as site-specific lipid fingerprints. This study highlights the role of abiotic conditions on the nutritional profile of *S. ramosissima* and establishes FA profiling as a potential avenue to trace the geographic origin of this halophyte plant. Overall, the present approach can make origin certification possible, safeguard quality, and enhance consumers’ trust in novel foods.

## 1. Introduction

Only 1% of the world’s plant species can develop under continuous exposure to high salt concentrations [1]. These plants are termed halophytes and have evolved to reproduce and thrive under saline environments, where NaCl concentration is equal to or greater than 200 mM and may, in extreme situations, peak at 400 mM [2,3]. In recent years, new applications have been found for halophytes, such as human nutrition [4], soil bioremediation [5,6], treatment of saline aquaculture effluents [7,8], or as a source of renewable energy such as biodiesel [9]. Increasing consumers’ demand for new and healthier foods, that can also be used as condiments (e.g., as an healthier alternative to salt) with improved nutritional properties, has promoted the consumption of halophytes [10].

With the increasing use of halophytes as food items, ensuring product quality, safety, and traceability, hinges on the development of markers which can successfully and reliably pinpoint the geographic origins of these plants [11]. Considering the increasingly extensive and complex supply chains of global markets, food traceability has become a prerequisite for food quality, safety, and public health [12,13]. Indeed, traceability is a legal requirement in many countries worldwide and is heavily regulated in the European Union [13,14]. Various methods have been used to determine the origin of foods, such as DNA profiling [15], elemental profiling [16], stable isotope analysis using 13C, 15N, and 18O [17] or lipid analysis, with a focus on fatty acid (FA) profiling [18,19]. The use of FA profiling for traceability has been tested in various plant matrices, such as olive oil [20], coffee beans [21], and nuts [22]. This method has proven effective in tracing the geographic origin of several plants and fruits [11,23]. According to the previous authors, polyunsaturated fatty acids (PUFAs) have been suggested as the most relevant FAs, contributing to location discrimination, with FAs C18:4*n*-3, C16:0, C22:0, and C18:3*n*-3 as the main contributors to the observed variability, depending on the type of plant or fruit. However, there are currently no studies on the traceability of halophytic plants.

Variations in FAs impact plant cell membranes, changing their fluidity and thereby improving their adaptation to external environmental conditions [24]. Thus, the FA profile can vary depending on the environmental conditions to which plants are subjected [25,26] and is primarily affected by extrinsic factors, such as temperature and salinity [26,27,28]. For instance, increased salinity and low temperature lead to a decrease in saturated fatty acid (SFA) levels, which stabilize the structure of the phospholipid bilayer, enhancing membrane fluidity [29]. Such variations in the FA profile of halophyte plants, particularly regarding essential FAs, such as the PUFAs alpha-linolenic acid (ALA, 18:3*n*-3) and linoleic acid (LA, 18:2*n*-6), enhance their nutritional value for different nutritional, biomedical, and biotechnological applications [30].

Among the numerous halophytes currently used for human consumption, *Salicornia ramosissima* stands out as one of the most produced and commercialized species and is well-established in various European markets [31,32]. Additionally, this plant is characterized by a nutritionally relevant lipidome, with the presence of *n*-3 FAs [31,33]. The increased utilization of this halophyte in different geographic areas has driven the need for its nutritional valorization, as well as the development of reliable traceability methods that may allow for the pinpointing of its geographic origin. Therefore, the present study aimed to characterize the FA profile of *S. ramosissima* originating from different locations along the coastline of mainland Portugal and evaluate the application of their FA profiles as a tool to trace their geographic origin.

## 2. Results

### 2.1. Fatty Acid Profiles

A total of 11 FAs were identified in the lipid extract of *S. ramosissima* from three locations (RAv, TE, and RF) (Table 1, Appendix A), while 12 FAs were identified in the shoots from plants sampled at ME (this included a FA on its epoxide form (9,10-epoxy-octadecanoic acid), with 18 carbons oxidized at positions 9 and 10) (Appendix A). All samples displayed a high content of long-chain FAs, with most of these biomolecules displaying chain lengths that ranged between 16 and 18 carbons (Table 1, Appendix A).

The most abundant FAs across all locations were C18:3*n*-3, C18:2*n*-6, and C16:0, with relative abundances ranging from 38.8% to 51.2%, 16.9% to 25.1%, and 18.1% to 20.3%, respectively, according to different sampling locations. In TE and RF, the content of these FAs decreased in the following sequence C18:3*n*-3 > C16:0 > C18:2*n*-6. However, in RAv and ME, the level of C18:2*n*-6 was higher than that of C16:0.

The most representative class of FAs in the studied plants was PUFA, including the essential FAs ALA (C18:3*n*-3) and LA (C18:2*n*-6). ALA was the most abundant FA in all locations, with its relative abundance ranging from 38.8% to 51.2%, peaking at RF (Appendix A). LA relative abundance varied from 16.9% (TE) to 25.1% (RAv). Both PUFA exhibited similar relative abundances in TE and RF, while these differed from that in ME and RAv.

RF presented the lowest *n*-6/*n*-3 ratio, whilst the highest ratio was recorded in ME (Figure 1). Ratio values in northern locations—RAv (0.55) and ME (0.58)—differed from those observed in southern locations—TE (0.34) and RF (0.35).

The plants sampled in all locations exhibited an AI and an IT equal to or lower than 0.35. The lowest AI values (Figure 2a) were observed in RAv (0.26) and RF (0.28), which differed significantly from AI in ME (0.35) and TE (0.32) (statistical outputs shown in Appendix A). Similarly, the lowest IT values (Figure 2b) were recorded in RAv (0.15) and RF (0.16) and were significantly different from those in ME (0.21) and TE (0.19).

### 2.2. Geographic Traceability

The ANOSIM performed to evaluate the similarity among FA profiles between locations yielded a global R of 0.686 (*p* = 0.001). On the one hand, the locations with higher R values were those found to be geographically further apart, such as RAv vs. RF (R = 0.87) and ME vs. RF (R = 0.895) (pairwise comparisons from ANOSIM are shown in Appendix A). On the other hand, although TE and RF, which are geographically closer, exhibited a lower R statistic (0.58), RAv and ME, also more closely located to each other, yielded a higher R value (0.697).

The CAP performed achieved a correct allocation of 93% of samples (Figure 3, Appendix A). Plants collected from RF exhibited the highest percentage of correct classifications (100%), followed by samples from TE (96.7%), RAv (90.0%), and ME (86.7%). CAP results showed an overall model success rate of 95% (Appendix A). When comparing the modelled predictions with the allocation of blind samples, the error for Rav and RF was 0%, i.e., all Rav and RF samples were correctly allocated to their places of origin. The errors associated with ME and TE were 40% and 27%, respectively.

## 3. Discussion

Halophytes, much like any other plants, are subject to various abiotic stress factors depending on their living environment; these include temperature, water, light, salt, and metal contamination, among others [26,34,35]. Most of these physicochemical factors vary geographically, leading to different environmental conditions that can influence plant development [25] and consequently alter nutrient absorption, photosynthetic activity, and metabolism [36,37]. The present study highlights how different geographic origins (with different environmental conditions) can shape the nutritional profile of *S. ramosissima* and provides evidence to support the use of FA profiles from their shoots to confirm the geographic origin of these halophyte plants.

The profiles recorded in *S. ramosissima* shoots were consistent with those previously described in the literature for the genus *Salicornia*, wherein C18:*n*-3, C18:2*n*-6, and C16:0 were indicated to be the most abundant FAs [33,38]. The data retrieved are in agreement with those reported by Maciel et al. [33] for *S. ramosissima* and Ventura et al. [38] for *Salicornia persica*. Further, a FA exclusive to ME, namely the oxidized FA 9,10-epoxy-octadecanoic acid, was identified through the presence of characteristic fragment ions in its MS, which showed the molecular ion (*m*/*z* 312) and the fragmented ion generated by the McLafferty rearrangement (*m*/*z* 74), as expected for a methyl ester derivative of a SFA [39]. However, the most useful fragment ions for identification were those at *m*/*z* 155 and *m*/*z* 199. The *m*/*z* 155 ion is formed from the terminal part of the molecule after cleavage between carbons 8 and 9. The fragment ion at *m*/*z* 199 is formed by the cleavage between carbons 10 and 11 [40]. This oxidized FA has been found in the seed oils of some plants [40,41] but, to the best of our knowledge, it has not been previously described in halophytes. Oxidized FAs constitute the cutin, the main component of the cuticle of several plants, which creates a protective film [42]. The formation of oxidized FA by plants is most likely a defense mechanism, which is triggered by both biotic and abiotic stressors [41]. Christie and Han [40] have also suggested that small amounts of epoxy FAs may form due to prolonged storage. Many FA oxidation reactions are catalyzed by cytochrome P450, one of the largest protein superfamilies in plants [41], which can be transcriptionally regulated by biotic and abiotic stressors, leading to the synthesis of oxidized FAs [43]. Thus, the presence of 9,10-epoxy-octadecanoic acid in ME may be associated with the exposure of the sampled plant to potentially stressful environmental conditions. It is unlikely that the recording of this epoxy FA could have been caused by a laboratory artefact, as only specimens sampled from ME consistently exhibited this biomolecule (it was present in all biological replicates from this specific location) and it was not recorded in any other plants sourced elsewhere, which were collected, stored, and processed under exactly the same conditions. However, given the novelty of this finding, a further analysis of the lipid profile of *S. ramossiima* sampled from different areas within the sampled habitat screened during the present study is required to confirm the prevalence of this epoxy FA.

The presence of long-chain unsaturated *n*-3 and *n*-6 FA enhances the nutritional value of halophytes [31]. The relative abundances recorded for ALA (*n*-3) and LA (*n*-6) were in the same order of magnitude as those previously reported in the literature for *S. ramosissima* (40% ALA and 20% LA; [33]), *S. persica* (48% ALA and 23% LA; [38]), *S. perennans* (20% LA; [44]), and *S. brachiata* (26% LA; [9]). Similarly, the results obtained in the northernmost locations (RAv, R = 0.55; ME, R = 0.58) are also consistent with the available literature [33], where *n*-6/*n*-3 ratios of 0.51 were reported for *S. ramosissima* sampled exactly in the same study area.

ALA and LA are precursors of anti-inflammatory and pro-inflammatory eicosanoid compounds, respectively [45]. These molecules must be consumed through diet as humans lack the enzymatic machinery to synthesize them de novo [46]. Indeed, an increased consumption of ALA is associated with higher tissue levels of ALA, EPA, and, to a lesser extent, DHA [47]. The latter two FAs are important precursors of eicosanoid molecules with a crucial role in the anti-inflammatory process [45,48]. Additionally, ALA can assist in reducing arachidonic acid (AA) levels by competing with LA for the same enzymes [49]. The conversion of LA into AA and ALA into EPA and DHA is achieved using the same set of enzymes [49], requiring competition between *n*-3 and *n*-6 FAs for desaturation enzymes. However, Δ4- and Δ6-desaturase enzymes have a higher affinity for *n*-3 FAs than for *n*-6 FAs. In this way, as an essential FA for the synthesis of anti-inflammatory compounds, ALA intake contributes to the prevention of coronary diseases [48]. Nevertheless, the consumption of *n*-6 FAs is also necessary for maintaining the body’s homeostasis [49,50]. It is therefore important to ensure a balanced consumption of these two families of FAs, with a predominance of *n*-3 FAs [47,48,50]. Indeed, low *n*-6/*n*-3 ratios, such as those found in TE and RF, are desirable in foods and reduce the risk of cardiovascular diseases and inflammatory conditions [50].

Both AI and TI are used to evaluate and predict the benefits associated with lipid intake, serving as indicators of cardiovascular risk, and thus allowing the inference of the lipid quality of foods [51]. AI reflects the relationship between SFAs, considered to be pro-atherogenic, and unsaturated FAs, either monounsaturated Fas (MUFAs) or PUFAs, considered to be anti-atherogenic. It evaluates the balance between Fas which favor lipid adhesion to immune and circulatory system cells and Fas which inhibit lipid plaque formation [51]. TI is defined as the ratio between SFAs (pro-thrombogenic) and MUFAs and *n*-3 and *n*-6 PUFAs (anti-thrombogenic), mirroring the trend to form blood vessel clots [51]. Therefore, low values (<1) of AI and TI are proxies of a better nutritional value of a given food [51]. The low values recorded for the shoots of *S. ramosissima* can be attributed to the high levels of ALA present in this plant. The locations RAv and RF presented the lowest values of these indices, and the results obtained were generally lower than those reported in the literature for *S. ramosissima* and other genera of halophyte plants [10]. The values for these indices (AI = 0.73 and TI = 0.36), reported by Barreira et al. [10], differed the most from those reported in the present study. The values reported by Ventura et al. [38] for other halophyte genera (e.g., *Arthrocnemum* and *Sarcocornia*) (AI: 0.45–0.62; TI: 0.21–0.37) are also higher than those recorded for *S. ramosissima* in RAv, ME, TE, and RF (AI: 0.23–0.42, TI: 0.12–0.26).

Geographic traceability can play a crucial role for producers aiming to enhance the value of their products, by safeguarding those in the value chain, namely consumers, and for assigning marketed plants originating from safe areas in terms of the presence of metals and other substances that might pose a threat to food safety and public health. In the present study, plants with similar R statistics also exhibited similar FA profiles, suggesting that they are likely exposed to similar environmental conditions due to their proximity. Moreover, in the Portuguese mainland coast, lower temperatures are recorded further north than in the south of coastal areas where halophyte plants occur [52]. Lower temperatures result in higher levels of FA unsaturation, allowing cell membranes to maintain their fluidity without compromising the homeostasis between intracellular and extracellular environments [53]. This acknowledged adaptation was confirmed in the present study, as samples of *S. ramosissima* collected from ecosystems located in northern mainland Portugal, characterized by lower temperatures, exhibited higher levels of unsaturation (RAv and ME: 20–22 °C, 74.4% and 65.2% unsaturated FA, respectively) when compared to ecosystems further south (TE and RF: 22–24 °C, 68.1% and 70.8% unsaturated FA, respectively). In addition to temperature, salinity can also shape the FA profile of halophyte plants, as a higher abundance of long-chain FAs and PUFAs can be related to the salt resistance of these organisms [54]. The results recorded in the present study confirm that *S. ramosissima* features high levels of long-chain FAs and PUFAs, with C18:2*n*-6 and C18:3*n*-3 playing a key role in the salt resistance displayed by halophytes. These plants have developed various mechanisms to regulate membrane fluidity by shifting their FA composition, such as by increasing the percentage of long-chain FAs [55], as documented in *Artemisia santonica* or *Salicornia perennans* [44], or by enhancing the individual contributions of FAs to the overall level of unsaturated FAs [44,54].

The success rate achieved by the CAP performed (93%) demonstrates that the FA profile of *S. ramosissima* shoots can be used, with a high degree of confidence, to successfully discriminate between different places of origin for these halophyte samples.

## 4. Materials and Methods

### 4.1. Sample Collection and Preparation

Samples of *S. ramosissima* were collected during the summer of 2019 from four different locations along the coast of mainland Portugal where this species is commercially explored: the Ria de Aveiro (Aveiro, RAv, n = 30); Mondego estuary (Figueira da Foz, ME, n = 30); Tagus estuary (Salinas do Samouco, Alcochete, TE, n = 30); and Ria Formosa (Faro, Algarve, RF, n = 27) (Figure 4). In RAv, ME, and TE, samples (plant shoots) were obtained from wild plants collected from former salt pans, where muddy soil prevails, while RF samples were kindly supplied by RiaFresh^®^ (Faro, Algarve), who use greenhouses to produce these halophytes in hydroponics at a commercial scale using a deep-water culture approach employing floating rafts. All samples were stored in aseptic plastic bags and brought to the laboratory under refrigerated conditions. Subsequently, plant shoots were washed, first with running water and then with distilled water, and stored at −80 °C until being freeze-dried and macerated using a mechanical mortar (RM 200; Retsch GmbH, Haan, Germany). The macerated biomass was once again stored at −80 °C until biochemical analysis.

### 4.2. Lipid Extraction

Total lipids were extracted from *S. ramosissima* samples following a method adapted from Bligh and Dyer [56]. Briefly, for each sample of *S. ramosissima*, 1 mL of CH_2_Cl_2_ and 2 mL of MeOH were added to a glass tube holding 50 mg of biomass. The mixture was vortexed for 1 min, followed by 1 min in an ultrasonic bath. The tubes were then incubated on ice, using an orbital shaker for 1 h. The mixture was centrifuged at 392× *g* for 10 min. The organic phase was collected into a new glass tube, washed with 1.2 mL of Milli-Q water (Synergy, Millipore Corporation, Billerica, MA, USA), and centrifuged again at 392× *g* for 5 min. The organic phase (lower layer) was transferred to a new glass tube and the sample was dried under a stream of nitrogen. Lastly, the lipid extract was dissolved in CH_2_Cl_2_, transferred to vials, dried under a stream of nitrogen, weighed, and stored at −20 °C until the FA profile was determined.

### 4.3. Fatty Acid Analysis

FA methyl esters (FAME) were obtained through alkaline transesterification [57]. Briefly, 60 μg of lipid extract in CH_2_Cl_2_ were transferred to a glass tube (previously washed with hexane) and the solvent was evaporated under a stream of nitrogen. Then, 1 mL of a solution of methylated internal standard C19:0 (1 μg mL^−1^; Sigma-Aldrich, St. Louis, MO, USA), prepared in hexane, and 200 μL of KOH (2M) in methanol were added to the tube. After vortex agitation for 2 min, 2 mL of an aqueous NaCl solution (10 g L^−1^) was added, followed by centrifugation for 5 min at 392× *g*. A volume of 600 μL of the organic phase was collected into an Eppendorf tube, and the FAME sample was concentrated under a stream of nitrogen for subsequent analysis by gas chromatography–mass spectrometry (GC-MS). FAME analysis was performed using an Agilent 8860 GC System coupled to an Agilent 5977B Network Mass Selective Detector equipped with an electron impact source operating at 70 eV and 250 °C and an Agilent 123-3232 DB-FFAP column (30 m × 320 μm × 0.25 μm) (all Agilent Technologies, Inc., Santa Clara, CA, USA). For GC-MS analysis, the derivatized extract was dissolved in 100 μL of hexane, transferred to an insert vial, and 2 μL of the sample were injected using a G 4513 A autosampler (Agilent Technologies, Inc.). Sample injection was performed in splitless mode, with a splitless time of 3 min; the injector was set at 220 °C and the detector at 250 °C. The temperature profile employed was as follows: 80 °C for 2 min; linear increase to 160 °C at 25 °C min^−1^; linear increase to 210 °C at 2 °C min^−1^; and linear increase to 225 °C at 20 °C min^−1^. This temperature was then maintained for 15 min. Helium was used as carrier gas at a constant flow rate of 1.4 mL min^−1^. Mass spectra were acquired in full scan mode within *m*/*z* 50–550.

### 4.4. Fatty Acid Identification and Integration

In the process of derivatization, FAs were methylated to enhance their volatility in GC-MS analysis. Thus, from the obtained GC-MS chromatograms, the corresponding FAME for each FA present in the sample was identified. FAME identification was performed via the Agilent MassHunter Qualitative 10.0 software, supported by the NIST14L library (Agilent Technologies, Inc.). Confirmation of FAME identification was also performed by comparing retention times and mass spectra with commercial FAME standards (Supelco 37 Component FAME Mix; Sigma-Aldrich). Agilent MassHunter Qualitative 10.0 software was used for peak integration corresponding to each FAME. The quantity of FAs was calculated as a relative percentage, using the area of each peak obtained through integration and normalized to the area of the internal standard (C19:0).

### 4.5. Lipid Quality Indices

The atherogenicity and thrombogenicity indices (AI and TI, respectively) used to estimate the lipid quality of foods for human health, were calculated using the following equations [58]:AI = (*C*12:0 + 4 × *C*14:0 + *C*16:0)/[∑*MUFA* + ∑(*n*-3 *PUFA*) + ∑(*n*-6 *PUFA*)]
TI = (*C*14:0 + *C*16:0 + *C*18:0)/[0.5 × ∑*MUFA* + 0.5 × ∑(*n*-6 *PUFA*) + 3· ∑(*n*-3 *PUFA*) + (∑*n*-3 *PUFA*/∑(*n*-6 *PUFA*)]

### 4.6. Statistical Analyses

The relative abundances of FAs in *S. ramosissima* shoots were initially standardized by log transformation and a similarity matrix between samples was obtained using the Bray–Curtis similarity coefficient. A one-way analysis of similarity (ANOSIM) was conducted to identify the existence of significant differences (*p* < 0.05) in the FA profiles of *S. ramosissima* shoots between the four locations sampled. For each FA, a one-way analysis of variance (ANOVA) was employed after checking for normality (Shapiro test) and variance homogeneity (Bartlett test) of residuals. Post hoc comparisons between locations were performed using Bonferroni tests whenever ANOVA revealed the existence of significant differences (*p* < 0.05). To evaluate the reliability of FA fingerprints of *S. ramosissima* to infer its geographic origin, a canonical analysis of principal coordinates (CAP) was performed. To validate the predictive model, half of the sample set was used for model development and the other half for blinded validation. Multivariate data analyses (ANOSIM and CAP) were performed using the Primer v7 + Permanova software (https://www.primer-e.com (accessed on 1 January 2020)), while ANOVA was performed using R software (v. 4.1.3) [59].

## 5. Conclusions

Abiotic factors, particularly temperature and salinity, significantly impact the FA profile of *S. ramosissima* and most likely drive latitudinal variation along the coastal areas of mainland Portugal. Across all locations, *n*-6/*n*-3 ratios, AI, and TI values were consistently below 1, highlighting the positive nutritional attributes of *S. ramosissima* for human health and improved nutritional value. The most appealing nutritional features for human consumption were recorded in plants from RF, suggesting that warmer climates may favor the nutritional profile of *S. ramosissima*. The profiling of fatty acids present in the shoots of this halophyte plant can allow the differentiation of specimens originating from different geographic locations, thus making it possible to enhance food safety by revealing whether plants originate from contaminated areas unsuitable for human consumption. Moreover, it may also reveal if plants are sourced from protected areas, where no-take zones are established. Future studies should aim to infer the maximum resolution that this approach may allow in terms of pinpointing geographic origins, as well as in the potential existence of seasonal and interannual variability.

## Figures and Tables

**Figure 1 plants-13-00545-f001:**
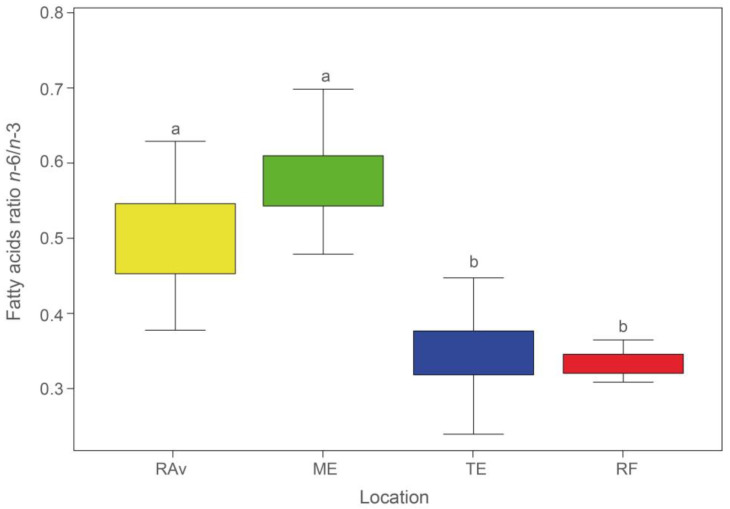
Average *n*-6/*n*-3 ratios (±SD) in *Salicornia ramosissima* shoots sampled from Ria de Aveiro (RAv), Mondego estuary (ME), Tagus estuary (TE), and Ria Formosa (RF). Different letters indicate significant differences between locations (ANOVA, *p* < 0.05).

**Figure 2 plants-13-00545-f002:**
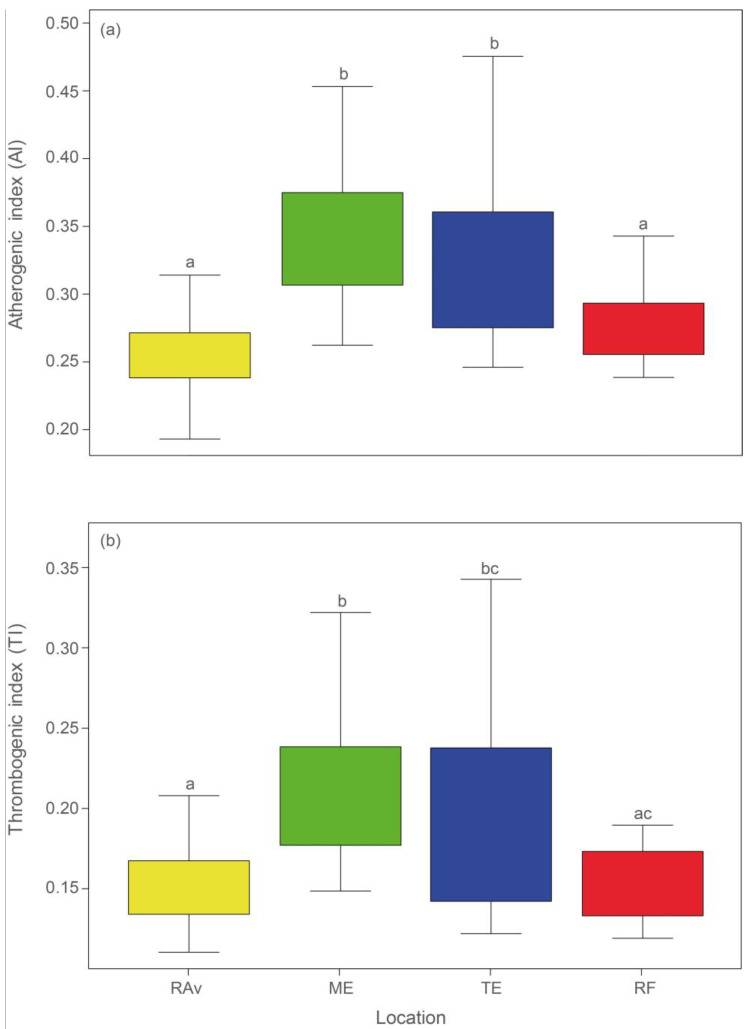
Atherogenicity index (IA) (**a**) and thrombogenicity index (IT) (**b**) recorded in *Salicornia ramosissima* shoots sampled from Ria de Aveiro (RAv), Mondego estuary (ME), Tagus estuary (TE), and Ria Formosa (RF). Different letters indicate significant differences (ANOVA, *p* < 0.05).

**Figure 3 plants-13-00545-f003:**
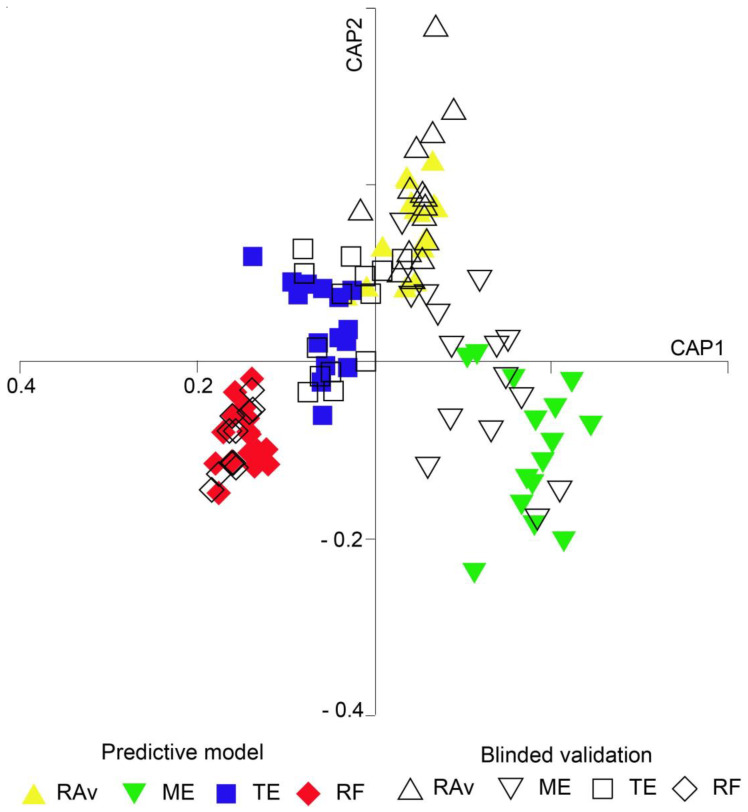
Canonical analysis of principal coordinates (CAP) based on fatty acid profiles of *Salicornia ramosissima* shoots sampled from Ria de Aveiro (RAv), Mondego estuary (ME), Tagus estuary (TE), and Ria Formosa (RF).

**Figure 4 plants-13-00545-f004:**
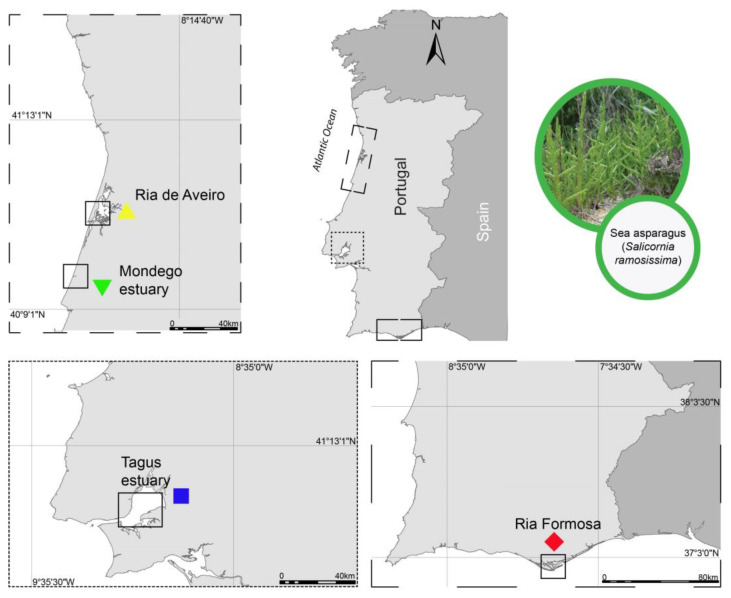
*Salicornia ramosissima* collection locations along the Portuguese mainland coast: Ria de Aveiro (RAv: 40.628094 N, 8.660742 W), Mondego estuary (ME: 40.112639 N, 8.832532 W), Tagus estuary (TE, 38.728075 N, 9.014552 W) and Ria Formosa (RF: 37.017829 N, 8.003740 W).

**Table 1 plants-13-00545-t001:** Relative abundance (%) of the fatty acids (FAs) present in the shoots of *Salicornia ramosissima* (mean values ± standard deviation) and their respective variation for each FA present in the four locations sampled: Ria de Aveiro (RAv, n = 30); Mondego estuary (ME, n = 30); Tagus estuary (TE, n = 30); and Ria Formosa (RF, n = 27).

FAs	RAv	ME	TE	RF
C14:0	0.3 ±0.07	0.6 ± 0.17	0.4 ± 0.07	0.2 ± 0.04
C16:0	18.1 ± 1.58	20.2 ± 1.7	20.3 ± 1.66	19.1 ± 1.09
C16:1	1.0 ± 0.13	0.6 ± 0.16	1.0 ± 0.13	1.6 ± 0.23
C17:0	0.3 ± 0.06	0.3 ± 0.10	0.4 ± 0.13	0.2 ± 0.08
C18:0	5.1 ± 2.43	6.8 ± 3.22	9.0 ± 4.32	7.1 ± 4.17
C18:1*n*-9	2.6 ± 0.48	3.4 ± 1.10	1.6 ± 0.37	0.9 ± 0.28
C18:2*n*-6	25.1 ± 2.61	22.5 ± 2.63	16.9 ± 1.71	17.1 ± 1.34
C18:3*n*-3	45.8 ± 3.65	38.8 ± 3.47	48.6 ± 5.54	51.2 ± 3.68
C20:0	0.5 ± 0.19	0.6 ± 0.15	0.5 ± 0.23	0.6 ± 0.09
C22:0	0.5 ± 0.26	1.0 ± 0.35	0.7 ± 0.30	0.6 ± 0.22
9,10-epoxy-octadecanoic acid		4.0 ± 3.57		
C24:0	0.8 ± 0.24	1.3 ± 0.53	0.7 ± 0.32	1.5 ± 0.38

## Data Availability

All raw data data of relative abundance (%) of fatty acids (FAs) analysis is available as Appendix A.

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
