# Peer review of "Fatty Acid Profiling as a Tool for Fostering the Traceability of the Halophyte Plant *Salicornia ramosissima* and Contributing to Its Nutritional Valorization"

_plants, 2024, doi:10.3390/plants13040545_

Round 1

Reviewer 1 Report

Comments and Suggestions for Authors

The article focuses on an interesting family of plants and investigates on the effect of geographic regions on the lipid content (qualitatively and quantitatively) of a common halophyte species.

I have doubts about the description of the 3 wild geographical sites (additional information on soil nature are needed) and of the hydroponic metod of cultivation (the growing media and cultivation method need to be reported). This aspect needs to be solved in order to make the article suitable for publication. Furthermore, the additional information requested will contribute to discussion.

Beside this, I have minor suggestions:

- figure 1 is not present in the manuscript (the first is figure 2)

- figure 2 and 3 (according to the present numbering) must be improved in quality and type of plot. I suggest to use a box plot to better show the variability among samples

- keywords must be different to woords used in the title

Author Response

Revision letter

Replies to each reviewer comment are provided bellow as RxRy (with x being the number of the reviewer and y the number of the comment by the same reviewer).

The anonymous reviewers are acknowledged, as their constructive criticism helped to improve the overall quality of the final manuscript.

Reviewer 1

R1C1: The article focuses on an interesting family of plants and investigates on the effect of geographic regions on the lipid content (qualitatively and quantitatively) of a common halophyte species.

I have doubts about the description of the 3 wild geographical sites (additional information on soil nature are needed) and of the hydroponic method of cultivation (the growing media and cultivation method need to be reported). This aspect needs to be solved in order to make the article suitable for publication. Furthermore, the additional information requested will contribute to discussion.

R1R1: The reviewer raises an important topic, but one must highlight that the aim of the present study was to characterize the FA profile of S. ramosissima originating from different locations along the coastline of mainland Portugal and evaluate the application of their FA profiles as a tool to trace their geographic origin. Considering the results achieved in this study we can say that yes, FA profiles of S. ramosissima can be used for such purpose - trace the geographic origin of these halophyte plants. Another issue is to shed light over the mechanisms that shape such FA fingerprints and to answer that question, detailed information on soil features and culture conditions on hydroponics will be paramount. In this way future studies being developed by our research group, aiming to shed light over the mechanisms shaping the FA profiles of halophyte plants in general, an S. ramosissima in particular, musty address these issues, analysing samples of sediment and hydroponic media for physicochemical and elemental analysis. In the present study, being a first approach to use this methodology and given the consistency of our findings, we consider that our study stands as it is. Moreover, as S. ramosissima samples were collected in 2019, any samples of soil or hydroponic media would not realistically reflect the physicochemical and elemental conditions of those same matrix at the time of sampling of the halophyte specimens used for FA analysis. Nonetheless, to best accommodate the suggestion made by Reviewer 1, we have added the following information in sub-section “4.1. Sample collection and preparation” in the “Material and Methods”, which now reads as follows:

“In RAv, ME, and TE samples (plant shoots) were obtained from wild plants collected from former salt pans, where muddy soil prevails, while in RF samples were kindly supplied by RiaFresh® (Faro, Algarve) that use greenhouses to produce these halophytes in hydroponics at a commercial scale using a deep-water culture approach employing floating rafts.”

R1C2: figure 1 is not present in the manuscript (the first is figure 2)

R1R2: We acknowledge Reviewer 1 for spotting this mistake in our original manuscript. Figures are now correctly numbered along the manuscript.

R1C3: figure 2 and 3 (according to the present numbering) must be improved in quality and type of plot. I suggest to use a box plot to better show the variability among samples

R1R3: Corrected as suggested by Reviewer 1.

R1C4: keywords must be different to words used in the title

R1R4: To best accommodate this important suggestion by Reviewer 1, which we acknowledge, Keywords on our revised manuscript were changed to: “GC-MS; geographic origin; healthy products; lipid markers; sea asparagus”.

Reviewer 2 Report

Comments and Suggestions for Authors

The manuscript ID 2851418 for "Fatty acid profiling as a tool to foster the traceability of the halophyte plant Salicornia ramosissima and contribute to its nutritional valorization" is well written with many useful information for the lipidomics community. It would be great to improve the manuscript with more information with the geographical location of sample collection and the reason for selecting those locations. 

Also please add more in the discussions, how to relate the fatty acid composition changes at the different location with the salt content in the soil and the temperature, where plants are grown.

Author Response

Revision letter

Replies to each reviewer comment are provided bellow as RxRy (with x being the number of the reviewer and y the number of the comment by the same reviewer).

The anonymous reviewers are acknowledged, as their constructive criticism helped to improve the overall quality of the final manuscript.

R2C1: The manuscript ID 2851418 for "Fatty acid profiling as a tool to foster the traceability of the halophyte plant Salicornia ramosissima and contribute to its nutritional valorization" is well written with many useful information for the lipidomics community. It would be great to improve the manuscript with more information with the geographical location of sample collection and the reason for selecting those locations.

R2R1:

To best accommodate this important remark by Reviewer 2, we have revised the subsection “4.1 Sample collection and preparation” in the “Material and Methods” (please also see above R1R1). It now reads as follows: “Samples of S. ramosissima were collected during the summer of 2019 from four different locations along the coast of mainland Portugal where this species is commercially explored: Ria de Aveiro (Aveiro, RAv, n = 30); Mondego estuary (Figueira da Foz, ME, n = 30); Tagus estuary (Salinas do Samouco, Alcochete, TE, n = 30); and Ria Formosa (Faro, Algarve, RF, n = 27) (Figure 4). In RAv, ME, and TE samples (plant shoots) were obtained from wild plants collected from former salt pans, where muddy soil prevails, while in RF samples were kindly supplied by RiaFresh® (Faro, Algarve) that use greenhouses to produce these halophytes in hydroponics at a commercial scale using a deep-water culture approach employing floating rafts.”. Please note that the exact location of the sample collection (geographic coordinates) are detailed in the caption of Figure 4.

R2C2: Also please add more in the discussions, how to relate the fatty acid composition changes at the different location with the salt content in the soil and the temperature, where plants are grown.

R2R2: Please see above our reply to R1C1.
